# Antibiotic Use, Healthcare-Associated Infections, and Antimicrobial Resistance in Intensive Care Unit of a Serbian Tertiary University Hospital, 2018–2024: An Ecological Analysis

**DOI:** 10.3390/antibiotics14111110

**Published:** 2025-11-04

**Authors:** Vesna Šuljagić, Vojislava Nešković, Duško Maksimović, Ivo Udovičić, Danijela Đurić-Petković, Nenad Perišić, Ivan Leković, Đorđe Taušan, Bojan Rakonjac, Katarina Vasiljević, Nemanja Rančić

**Affiliations:** 1Department of Healthcare-Related Infection Control, Military Medical Academy, 11000 Belgrade, Serbia; 2Medical Faculty, Military Medical Academy, University of Defence, 11000 Belgrade, Serbia; vojkan43@gmail.com (V.N.); ivoudo@gmail.com (I.U.); danijeladjuricpetkovic@gmail.com (D.Đ.-P.); ivan.lekovic@gmail.com (I.L.); katarinalojanica@gmail.com (K.V.); nece84@hotmail.com (N.R.); 3Clinic for Anesthesiology and Critical Care, Military Medical Academy, 11000 Belgrade, Serbia; dumaksa06@gmail.com; 4Institute of Medical Microbiology, Military Medical Academy, 11000 Belgrade, Serbia; bonatejo@gmail.com; 5Clinic for Gastroenterology and Hepatology, Military Medical Academy, 11000 Belgrade, Serbia; gastronesa1@gmail.com; 6Clinic for Vascular and Endovascular Surgery, Military Medical Academy, 11000 Belgrade, Serbia; 7Treatment Sector, Military Medical Academy, 11000 Belgrade, Serbia; tausandjordje@gmail.com; 8Center for Clinical Pharmacology, Military Medical Academy, 11000 Belgrade, Serbia

**Keywords:** healthcare-associated infections, antibiotic consumption, antimicrobial resistance, intensive care unit, WHO AWaRe classification, *Clostridioides difficile*, Serbia

## Abstract

**Background**: Healthcare-associated infections (HAIs), antimicrobial resistance (AMR), and antibiotic use (AU) remain critical challenges in intensive care units (ICUs). Reliable long-term surveillance is essential to inform stewardship programs and infection prevention. **Methods**: We conducted a seven-year ecological study (2018–2024) of adult patients admitted to the surgical ICU of a Serbian tertiary university hospital. Patients with ICU stays >48 h were included. Data on demographics, HAIs, AU expressed as ‘days of therapy’ (DOT) per 100 patient-days, and resistance profiles of key pathogens were collected. AU was classified by WHO Access, Watch and Reserve categories. Trends were analyzed, and correlations between AU and healthcare-associated *Clostridioides difficile* infection (HA-CDI) incidence density as well as changes in AMR rates were assessed. Direct expenditures for antibiotic therapy were also calculated. **Results:** Among 2055 patients, 511 (24.9%) developed at least one HAI. HA-CDI showed a marked upward trend. Overall, AU was stable, but Reserve antibiotics increased significantly (R^2^ = 0.456), particularly linezolid, colistin, and ceftazidime–avibactam. Resistance to carbapenems and colistin in *Klebsiella pneumoniae* (*K. pneumonia*) demonstrated alarming trends. A strong correlation was observed between colistin use and colistin resistance in *K. pneumoniae*. Antibiotic expenditures peaked in 2023 at more than double pre-COVID levels, mainly driven by Reserve agents. **Conclusions:** This study reveals a concerning rise in multidrug-resistant pathogens, escalating Reserve antibiotic use, and substantial financial burden. Strengthened antimicrobial stewardship, optimization of Access, restriction of Reserve agents, and investment in infection prevention and surveillance are urgently needed.

## 1. Introduction

Numerous studies from various regions worldwide have shown that patients requiring intensive care are at a substantially higher risk of acquiring healthcare-associated infections (HAIs), owing to both intrinsic (e.g., underlying disease, age, immunosuppression) and extrinsic (e.g., mechanical ventilation, central venous catheter, urinary catheter) risk factors [1,2,3,4,5]. In such circumstances, the use of antibiotics for the treatment and prevention of HAIs is often irrational, which in turn contributes to additional problems, including the emergence of hospital-acquired *Clostridioides difficile* infection (HA-CDI) [6,7] and antimicrobial resistance (AMR) [8,9]. Drug-resistant HAIs in vulnerable patients—such as neonates, older adults, and those with chronic illnesses—complicate treatment, worsen survival outcomes, prolong hospital stays, and increase healthcare costs. The estimated all-age death rate attributable to resistance is highest in the least developed regions of the world [10]. Treatment of HAIs caused by multidrug-resistant Gram-negative bacteria has become an increasing challenge for patients in intensive care units (ICUs) [11,12].

This already complex situation was further exacerbated by the COVID-19 pandemic, which significantly disrupted healthcare systems worldwide. Meta-analyses report diverse patterns of antibiotic use (AU) in hospitals during the pandemic: some countries, after an initial surge in use early in the pandemic, later registered decreases, while others reported sustained increases. Importantly, standard antibiotic stewardship programs (ASPs) were often discontinued during this period [13,14]. Irrational antibiotic use carries substantial hidden and direct costs—from the micro-level (e.g., per prescription in hospitals) [15] to large-scale societal and economic impacts worldwide [16]. Evidence consistently demonstrates that targeted interventions—such as stewardship programs, checklists, strengthened hygiene measures, and clinician behavior nudges—are not only cost-effective but frequently cost-saving [17,18]. Given these challenges, every hospital requires a surveillance system to monitor HAIs, AU, and resistant microorganisms, as well as to evaluate the effectiveness of interventions aimed at preventing HAIs. In this study, we sought to evaluate the incidence density (ID) of specific HAIs, and to examine the relationship between AU at the patient level with ID of HA-CDI and changes in AMR rates over the past seven years in a cohort of surgical ICUs within a tertiary university hospital in Serbia. In addition, we assessed the economic burden of AU in the ICU.

## 2. Results

Out of 2055 patients who stayed in the surgical ICU for more than two days, 511 (24.9%) developed at least one HAI. The characteristics of these patients, stratified by study year, are presented in Table 1. The full list of ICD-10 diagnoses can be found in Appendix A.

When stratified by HAI status, in-hospital mortality was considerably higher among patients with HAI (60.1%) than among those without HAIs (41.0%).

Average ICU length of stay in days during 2018–2024 are shown in Figure 1.

Of the 343 cases of pneumonia reported, 98.0% were associated with intubation. Among patients who stayed in the ICU for more than two days, 16.7% experienced at least one episode of pneumonia. The mean ID of pneumonia was 14.1 episodes per 1000 patient-days, and when expressed per MV days, the mean rate of ventilator-associated pneumonia (VAP) was 20.0 per 1000 MV-days (Figure 2). Although the data show a decline in pneumonia incidence, it is unlikely that this fully reflects a true decrease in disease burden. Changes in diagnostic criteria, admission practices, or reporting may have influenced these results.

A total of 130 cases of ICU-acquired bloodstream infection (BSI) were reported. On average, ICU-acquired BSIs occurred in 6.3% of patients who remained in the ICU for more than two days. The mean ID of ICU-acquired BSI was 5.1 episodes per 1000 patient-days. Additionally, 52 central venous catheter (CVC)-related BSIs were recorded (during 23,846 CVC days), corresponding to a rate of 2.3 per 1000 CVC days.

A total of 122 cases of ICU-acquired urinary tract infection (UTI) were identified. On average, ICU-acquired UTIs occurred in 5.9% of patients hospitalized in the ICU for more than two days, with 100.0% of UTI episodes associated with the use of a urinary catheter (UC). The mean ID of ICU-acquired UTI was 4.9 episodes per 1000 patient-days, and when expressed per urinary catheter days, the mean rate of catheter-associated urinary tract infection (CAUTI) was 5.1 per 1000 UC-days.

The trend of HA-CDI ID is shown in Figure 3.

The ID of HA-CDI was lowest in 2018 (2.86 per 10,000 patient-days) and highest in 2024 (36.6 per 10,000 patient-days). A clear upward trend in HA-CDI ID was observed over time, with an average increase of 4.44 units per year. The model explained 72.24% of the variation (R^2^ = 0.72), indicating a strong time-related correlation.

Trends in antibiotic utilization over the 7-year period are summarized in Table 2. The statistical analysis of total AU and the distribution of antibiotics according to the World Health Organization (WHO) Access, Watch, and Reserve (AWaRe) classification during the study period is presented in Table 2.

Total utilization of antibacterials for systemic use (ATC J01 group) remained relatively stable, with peak utilization in 2023 (242.39 DOT/100 patient-days) and 2024 (242.03 DOT/100 patient-days), and the lowest utilization in 2021 (190.60 DOT/100 patient-days). No statistically significant overall change was detected (*p* = 1.000).

Several antibiotics (e.g., ampicillin, penicillin G, chloramphenicol, and erythromycin) demonstrated zero or negligible use and were excluded from the trend analysis.

Notable increasing trends were observed for: linezolid sharp increase from 1.6 in 2020 to 12.2 in 2024; colistin particularly rising from 2022 (7.8) to 2024 (20.1); ceftazidime–avibactam introduced in 2023, with continued growth into 2024; clindamycin steady increase, especially after 2020 (*p* = 0.057).

Notable decreasing trends were observed for: metronidazole significant decline from 62.6 in 2020 to 37.2 in 2024 (*p* = 0.002); imipenem/cilastatin downward trend from 20.7 in 2018 to 11.5 in 2024 (*p* = 0.023); levofloxacin decline from 5.6 in 2018 to 2.3 in 2024 (*p* = 0.023); tigecycline decreased from 2.37 in 2020 to 0.5 in 2024 (*p* = 0.057).

Vancomycin showed consistently high levels of use between 2018 and 2024 (28.7–38.2 DOT/100 patient-days). While year-to-year fluctuations were observed, no significant overall trend was detected (*p* = 0.447). Stable or fluctuating usage without clear trends was also observed for meropenem which peaked in 2024 (34.3, *p* = 0.704), and for piperacillin–tazobactam (*p* = 0.447). Ceftriaxone use fluctuated throughout the study period, with a pronounced spike in 2019 (24.4 DOT/100 patient-days). Despite these irregularities, a gradual and consistent increase was observed from 2020 onwards, reaching nearly 20 DOT/100 patient-days in 2024—more than double the 2018 value. The trend approached statistical significance (*p* = 0.057). Ceftazidime use demonstrated a significant downward trend, decreasing from 1.60 to 0.26 DOT/100 patient-days, a reduction that reached statistical significance (*p* = 0.002).

During the study period, the use of Access antibiotics showed a moderate decline (R^2^ = 0.514), while Watch antibiotics exhibited a slight and inconsistent decrease (R^2^ = 0.2212). In contrast, Reserve antibiotics demonstrated a clear upward trend (R^2^ = 0.4556) (Figure 4).

The correlation between the consumption of individual antibiotics and the ID of HA-CDI over the 7-year period is presented in Table 3. Notably, consumption of ceftazidime showed a statistically significant negative association with the ID of HA-CDI (r = –0.893, *p* = 0.007).

*Klebsiella pneumoniae* (*K. pneumoniae*) demonstrated critical resistance across nearly all major antibiotic groups, with particularly concerning recent increases in colistin resistance. Resistance to third-generation cephalosporins remained very high, fluctuating mostly between 90 and 100%, and reaching 100% in both 2020 and 2024. The upward trend in carbapenem resistance was nearly statistically significant (*p* = 0.057), rising from 40% in 2018 to 95.65% in 2024, despite a temporary decline to 77% in 2023. Resistance to colistin showed a statistically significant upward trend: while no resistance was observed until 2021 (0%), a sharp increase was noted thereafter—42.9% in 2022, 56.8% in 2023, and 47.8% in 2024 (Table 4).

*Providencia stuartii* exhibited substantial β-lactam resistance, with extremely high resistance rates to third-generation cephalosporins (60–100%), although a slight decrease was observed in 2024 (71.4%). Resistance to carbapenems showed variability but an overall increasing trend, rising from 0% in 2018 to 71.4% in 2024.

*Acinetobacter* spp. displayed persistently very high resistance to carbapenems (89–100%) throughout the study period. In contrast, resistance to colistin declined markedly, from 30% in 2018 to 0% between 2020 and 2022, with a slight rebound in 2023 (12.5%), followed again by 0% in 2024.

*Staphylococcus aureus* and *Enterococcus* spp. showed no consistent trends in resistance to oxacillin and vancomycin, respectively.

The correlation between the consumption of specific antibiotics and selected antimicrobial resistance over the 7-year period is presented in Table 5. A strong positive relationship was observed between colistin use and increased resistance of *K. pneumoniae* to colistin (r = 0.714), although the *p*-value indicated that this association did not reach statistical difference (*p* = 0.071).

Figure 5 illustrates temporal trends in ICU antibiotic consumption by WHO AWaRe classification (Access, Watch, and Reserve groups) alongside carbapenem resistance in *K. pneumoniae*, *Pseudomonas aeruginosa*, and *Acinetobacter* spp., colistin resistance in *K. pneumoniae*, as well as *Clostridioides difficile* infection (CDI) incidence during 2018–2024.

An analysis of antibiotic consumption costs is shown in Figure 6. Total expenditure on antibiotics peaked in 2023 at RSD 40,964,942, representing a 2.11-fold increase compared with pre-COVID 2019 levels (RSD 19,438,637). The largest increase was attributable to spending on Reserve antibiotics, which rose from RSD 3,315,800 in 2019 to RSD 15,777,189 in 2023—a 4.75-fold increase. A similar upward pattern was noted in the Watch group. In contrast, consumption of Access antibiotics remained relatively stable without significant temporal variation. A slight decline in overall antibiotic consumption was recorded in 2024. When AU in our ICU was expressed relative to the total health expenditure in Serbia, antibiotics accounted for approximately 0.005–0.007% during the investigation period.

## 3. Discussion

This descriptive study represents the largest analysis of HAIs, AU and AMR conducted in a Serbian surgical ICU over a 7-year period, including the COVID-19 pandemic years. Results have indicated that overall AU remained stable during study period, but there was a significant increase in the use of Reserve antibiotics—especially linezolid, colistin, and ceftazidime–avibactam—while the use of some Watch antibiotics declined. A strong correlation was observed between increased colistin use and colistin resistance in *K. pneumoniae*. Resistance to carbapenems and colistin in *K. pneumoniae* showed upward trends. Antibiotic expenditures peaked in 2023 at more than double pre-COVID levels, primarily driven by increased use of Reserve agents. The findings strongly emphasize the urgent need for improved ASP.

Following the first national campaign for rational AU in Serbia in 2015, launched by the Ministry of Health, a significant decrease in AU was observed in 2016 and 2017, with a total reduction of 32.79% compared with 2015 [19]. However, during the subsequent five-year period (2017–2021), which overlapped with the COVID-19 pandemic, AU increased markedly, particularly in 2020 and 2021. Wholesale data from Serbia similarly revealed a statistically significant upward trend, with a 16% increase in the ratio of broad-spectrum to narrow-spectrum antibiotic use during that time [20]. Previous national data on AU in healthcare settings remain limited [21,22,23], and information specifically concerning ICUs in Serbia is especially scarce [8].

In our surgical ICU, total utilization of systemic antibacterials remained relatively stable across the study years (*p* = 1.000), without a statistically significant long-term change. The lowest use was recorded in 2021 (190.60 DOT/100 patient-days) and 2022 (190.28 DOT/100 patient-days). By comparison, markedly lower AU has been reported in mixed adult ICUs at Ghent University Hospital, Belgium (1232 DOT/1000 patient-days) [24], and in northern Sweden (1177 DOTs in secondary care and 1261 DOTs in tertiary care ICUs) [25]. Automated monitoring in Sweden further showed that 66% of ICU patients received at least one antibiotic during admission, rising to 90% when the stay exceeded 48 h. Patients exposed to more than three antibiotic classes had a 30-day mortality rate of 40.6% [25]. In contrast, 98% of our patients received antibiotics, with overall hospital mortality ranging from 39–51%.

Interestingly, the lowest AU in our ICU occurred during the peak of the COVID-19 pandemic in Serbia, when Military Medical Academy (MMA) was designated to provide all specialist services except for COVID-19 care. At that time, the multidisciplinary antimicrobial team was inactive due to the limited availability of infectious disease specialists, who were reassigned to COVID-19 hospitals. When the specialists returned and stewardship activities resumed, AU returned to pre-pandemic levels. These observations align with findings by Macheda et al., who reported that only one in three infectious disease specialists worldwide recommended short-course antibiotic regimens [26]. Despite strong evidence from randomized controlled trials showing that short-term therapy is as effective as longer courses, with fewer side effects and lower resistance risks [27,28,29,30], such practices have not yet been fully implemented in our ICU.

This observation raises a hypothesis: in the absence of active stewardship oversight, ICU clinicians may have adopted more conservative antibiotic prescribing behaviors, possibly influenced by heightened infection prevention and control (IPC) measures, altered patient mix, or limited perceived indications. Alternatively, the pandemic context may have shifted focus toward non-COVID infection vigilance, inadvertently reducing unnecessary prescribing. These insights suggest that antibiotic prescribing in critical care may be shaped not only by clinical protocols and stewardship oversight, but also by institutional culture, education, and ICU-specific habits. Further research should explore whether targeted educational interventions or decision-support tools could bridge the gap between evidence and practice, especially in the use of short-course antibiotic therapies.

According to WHO, Access antibiotics should form the backbone of empirical therapy due to their lower cost and reduced potential to drive resistance. However, in our ICU, their use was limited and declined moderately but consistently over time (Y = –2.33x + 74.41, R^2^ = 0.51). Use was largely confined to aminoglycosides (amikacin 6–11 DOT; gentamicin 0.4–2.4 DOT) and trimethoprim–sulfamethoxazole (stable, low use 1.2–3.6 DOT), with minimal contributions from cefazolin (≤1.8 DOT). Clindamycin was the only Access antibiotic showing a steady upward trend, particularly from 2023 to 2024, though without statistical significance (*p* = 0.057). Similar findings were reported in Romania, where lincosamides were commonly prescribed for HAIs [31]. The progressive shift away from Access antibiotics toward broad-spectrum Watch and Reserve agents reflects the growing presence of multidrug-resistant (MDR) pathogens in our ICU, consistent with earlier reports [32]. This often resulted in the use of Reserve antibiotics as the first-line treatment.

Carbapenems represented one of the most frequently used antibiotic groups in our ICU. Their use remained consistently high, ranging from 41 to 54 DOT annually, without a clear overall trend. Imipenem/cilastatin declined significantly, while meropenem remained dominant with considerable variability. Ertapenem was occasionally used but declined in recent years. Compared to European Centre for Disease Prevention and Control (ECDC) data, which reported much lower carbapenem use in ICUs between 2019 and 2021 (maximum 39 DOT in Estonia) [1,33,34], our levels were substantially higher. Similarly, high reliance on carbapenems has been documented in ICUs in Saudi Arabia [35]. Prolonged carbapenem use has been associated with excess risk in surgical patients, underlining the importance of stewardship interventions [36]. Almost a quarter of the total number of patients (24.6%) included in our study, often at an advanced age, with numerous co-morbidities, were exposed to surgical intervention due to emergency conditions caused by mesenteric ischemia and consequent intestinal gangrene, duodenal perforation, etc. Overall, the persistent high reliance on carbapenems highlights their importance in treating severe infections in our patients but also raises concerns about the selective pressure of carbapenem-resistant organisms such as *Klebsiella*, *Acinetobacter*, and *Pseudomonas*.

In our ICU, glycopeptide use was overwhelmingly dominated by vancomycin, with consistently high levels ranging from 28 to 38 DOT per 100 patient-days. Teicoplanin was administered only in 2018 and subsequently disappeared from use. Overall glycopeptide consumption remained stable throughout the study period, reflecting the persistent clinical burden of resistant Gram-positive infections, particularly methicillin-resistant *Staphylococcus aureus* (MRSA) and vancomycin-resistant *Enterococcus* (VRE).

According to ECDC data, glycopeptide use in our ICU was markedly higher than the average reported across European ICUs. Specifically, in 2019, mean consumption was 4.9 DOT per 100 patient-days, with the highest recorded use in Portugal (9.4 DOT) [33]. Portugal again led glycopeptide consumption in 2020, with 8.9 DOT per 100 patient-days [34], while in 2021 Estonia reported the highest usage in Europe, at 11.2 DOT (1). Comparable findings have been reported outside Europe. For example, in King Abdulaziz Medical City, Riyadh, Saudi Arabia, vancomycin ranked as the third most frequently consumed antimicrobial across all ICUs, except in the burn ICU where other agents predominated [35].

A recent study demonstrated that the implementation of a dedicated ASP in a pediatric cardiac ICU did not result in increased mortality or greater use of alternative broad-spectrum antimicrobials. Instead, it achieved a sustained reduction in meropenem and vancomycin utilization [37]. Such findings underscore the potential for improved treatment quality in highly vulnerable groups, particularly pediatric patients, and indicate that the use of Watch group antibiotics can be effectively controlled. Similarly, a systematic review by Lindsay et al. showed that ASPs incorporating audit and feedback did not increase ICU mortality, supporting the safety of such programs with respect to patient outcomes [38].

The use of Reserve antibiotics was relatively moderate in the period 2018–2022 (10–17 DOT). However, since 2023, their consumption has tripled compared to earlier years (33–36 DOT per 100 patient days). Their use has sharply increased, driven mainly by rising consumption of linezolid and colistin, and the introduction of ceftazidime–avibactam. Colistin use showed a marked escalation: after moderate levels in 2018–2022 (7–13 DOT), its application rose to approximately 20 DOT in 2023–2024. By comparison, ECDC data for 2019–2021 reported an average polymyxin use of just 1.1 DOT across European ICUs, with the highest levels observed in spinal ICUs in Italy (4–5 DOT per 100 patient-days) [1,33,34].

Polymyxins, including colistin and polymyxin B, were introduced into clinical practice in the 1950s due to their broad activity against Gram-negative bacteria, but their use rapidly declined after frequent reports of nephrotoxicity and neurotoxicity two decades later [39]. At the start of the 21st century, the global spread of MDR Gram-negative pathogens led to renewed interest in these agents, and subsequent studies suggested that earlier concerns regarding colistin toxicity may have been overstated [40]. Nevertheless, contemporary data confirm substantial risks: nephrotoxicity rates assessed by internationally recognized criteria reached 39% among patients treated with systemic polymyxins, with 13% experiencing severe nephrotoxicity [41]. A meta-analysis by Wang et al. further estimated the pooled incidence of polymyxin-induced nephrotoxicity among ICU patients at 34.8%, highlighting the urgent need for enhanced monitoring and management strategies to mitigate adverse outcomes in this population [42].

Diarrhea is common among critically ill patients, though CDI accounts for only a minority of cases (≈9.3%) [43]. Meta-analyses estimated an ICU CDI ID of 3.24 per 10,000 patient-days in period 2016–2024 [44]. In our ICU, however, a statistically significant and marked upward trend was observed, culminating in 2024 with an ID of 36.64 per 10,000 patient-days, indicating an outbreak during that year [45]. Similar increases have been reported by UK surveillance systems across all age groups and sexes between financial year (FY) 2020 to 2021 and FY 2023 to 2024 [46]. Chen et al. estimated that over the past 30 years, the global age-standardized mortality rate due to CDI has increased dramatically from 0.19 in 1990 to 0.43 in 2019, particularly in regions with high to medium sociodemographic indices, as well as in populations of women and those aged 67.5 years and older. They also calculated that AU had a significant positive correlation with different risk stratifications (clindamycin had the highest risk among all antibiotics included, followed by carbapenems) [47]. In our study, clindamycin consumption showed a moderate but non-significant positive correlation with CDI, while ceftazidime demonstrated a significant negative association (r = −0.893, *p* = 0.007).

When compared with data from European and international surveillance systems, the ID observed in our ICU were within the expected range for high-acuity surgical ICU (Table 6).

ID of ICU-acquired pneumonia in our study was 14.1 per 1000 patient-days, with a VAP rate of 20.0 per 1000 MV-days. Both indicators were markedly higher than those reported by the ECDC (mean 6.8 per 1000 patient-days, 2019–2021) and also exceeded the pooled INICC rate (11.96 per 1000 MV-days). These findings likely reflect the high proportion of patients requiring prolonged MV and complex postoperative management in our surgical ICU.

The mean ID of ICU-acquired bloodstream infections (BSI) was 5.1 per 1000 patient-days, comparable to the ECDC mean (5.1 per 1000 patient-days). When expressed per device-days, our CLABSI rate was 2.3 per 1000 CVC-days, lower than the INICC pooled CLABSI rate (4.55 per 1000 central line-days). While this finding suggests effective implementation of central line care bundles and aseptic insertion techniques, alternative explanations should be considered. Differences in patient case mix or reduced central line dwell times could influence observed rates. Further, intensified IPC measures during the COVID-19 pandemic may have broadly suppressed CVC-related infections. These factors highlight the need for cautious interpretation and comprehensive surveillance practices.

For urinary tract infections (UTI), the mean ID was 4.9 per 1000 patient-days, while the CAUTI rate reached 5.1 per 1000 catheter-days. These values are higher than the ECDC mean (2.9 per 1000 patient-days) but remain close to the INICC pooled CAUTI rate (2.91 per 1000 catheter-days). The near-universal catheter use among UTI cases in our cohort underscores the importance of continued adherence to catheter management protocols and early removal policies.

When interpreting these comparisons, it is essential to recognize differences in denominators between surveillance systems: ECDC expresses rates per 1000 patient-days, while INICC uses 1000 device-days. Despite these methodological differences, our data consistently demonstrate a relatively higher burden of HAI, particularly pneumonia, likely reflecting the severity and complexity of cases in a tertiary surgical ICU setting.

In-hospital mortality was notably higher among patients with HAI compared to those without infection (60.1% vs. 41.0%) (Table 1). This finding is consistent with previous studies showing that HAIs significantly increase the risk of death and prolong ICU stay [48]. These results underline the importance of infection prevention and antimicrobial stewardship in improving patient outcomes.

WHO has identified AMR as a critical global public health threat [49]. Between 2010 and 2018, hospital-sector surveillance revealed high consumption of Reserve antibiotics in several Southern European countries, with a marked upward trend in polymyxin use observed in Denmark, Greece, Hungary, Italy, Portugal, Slovakia, and the United Kingdom. In these settings, consumption increased by more than 50% during the study period. At the broader EU/EEA/UK level, colistin use quadrupled, coinciding with a rising trend of resistance in *K. pneumoniae*, although not in other Gram-negative species [50].

A 2021 study further demonstrated that in most European countries providing data on ICU antimicrobial use (including Estonia, France, Germany, Hungary, Italy, Lithuania, Portugal, and Spain), prescriptions remained predominantly empirical rather than targeted [1]. For the EU overall, a statistically significant upward trend in AU was documented between 2019 and 2023. Resistance rates were consistently higher in Southern and Eastern Europe, particularly for *Escherichia coli* and *K. pneumoniae*, with Balkan countries—based on limited available data—ranking among the highest in Europe [51].

Carbapenem and colistin resistance in *K. pneumoniae* represent two of the most pressing clinical challenges in contemporary medicine. According to the latest joint ECDC/WHO report (data from 2021), carbapenem resistance exceeded 50% in *K. pneumoniae* isolates in 9 of 44 European countries, with the highest prevalence observed in Southern and Eastern Europe [51]. National data from Serbia are even more alarming: a multicenter study conducted in 2021–2022 reported resistance rates of 62.5% to imipenem and 71.3% to meropenem [52]. Consistent with these findings, our study revealed a sharp increase in carbapenem resistance from 40% in 2018, to 77% in 2023, and up to 95.7% in 2024. Although this upward trend was evident, statistical significance was narrowly missed (*p* = 0.057).

Uzairue et al. reported a pooled prevalence of colistin-resistant *K. pneumoniae* of 3.1% (95% CI: 1.5–4.7%), with markedly higher prevalence among isolates studied from 2020 onwards (12.9%). The highest rates were documented in Thailand (19.2%), while the lowest were observed in South Korea (0.8%). Significant colistin resistance has also been reported among ICU isolates [53]. In our cohort, resistance of *K. pneumoniae* to key antibiotics increased substantially, particularly with the critical rise in colistin resistance observed between 2022 and 2024 (42.9% → 56.8% → 47.8%). These results, however, should be interpreted cautiously, as only phenotypic resistance was assessed. Accurate evaluation requires genotypic testing to detect resistance genes. Beyond antibiotic consumption and hospital-specific factors, AMR is also influenced by molecular and genetic mechanisms, biofilm formation, and broader pharmacological, ecological, and epidemiological determinants. Nevertheless, our findings reinforce the urgent need for ongoing surveillance of MDR strains.

The conclusions presented in this study are based on phenotypic resistance testing. While molecular typing was not routinely conducted for all isolates, phenotypic carbapenemase detection was performed using the NG-Test CARBA 5 assay since 2021. In our prior molecular investigation [54], PCR characterization revealed that the predominant carbapenemase gene among *K. pneumoniae* isolates was OXA-48-like (67.3%), followed by NDM (24.7%) and KPC (18.3%), with VIM detected in 1.6% and IMP not detected. These genotypic data strengthen the interpretation of our phenotypic findings and confirm that resistance in our ICU is largely driven by well-known carbapenemase mechanisms. Future work should expand molecular surveillance to capture emerging variants and transmission patterns in ICU settings.

The financial cost related to antimicrobial treatment is one of the main reasons of the highest health economic burden in ICU. In our study antibiotic expenditure peaked in 2023, driven largely by increased spending on Reserve and Watch antibiotics. This pattern mirrors findings from other studies [55,56], which consistently identify ICUs as accounting for nearly half of total hospital antibiotic expenditures [57]. Our data reinforce the need for structured ASP to optimize antibiotic use while reducing costs. Increasing reliance on Access antibiotics and restricting Reserve/Watch use to the most severe cases could not only reduce resistance pressures but also redirect financial resources toward essential infection control measures, diagnostic technologies, and staff protection. This would have the goal of shorter patient stays in ICUs, faster recovery and a lower mortality rate.

During the seven-year study period, when AU in our ICU was expressed relative to the total health expenditure in Serbia, antibiotics accounted for approximately 0.005–0.007%. When viewed in the broader national healthcare context, AU in our ICU represents only a minor fraction (<1%) of total health expenditures. Nevertheless, even small proportional reductions achieved through stewardship interventions can yield meaningful cost savings. Evidence from multiple studies demonstrates that ASPs in ICUs can reduce antibiotic expenditures by 20–35% without negatively affecting patient outcomes [58,59]. Therefore, beyond their roles in infection control and resistance prevention, stewardship initiatives directly contribute to resource optimization and financial sustainability. Implementing targeted ASP strategies—such as antibiotic time-outs, daily therapy review, and de-escalation protocols—offers both cost-containment benefits and improved patient safety.

This study was based on aggregated (ecological) surveillance data rather than patient-level information. Therefore, the associations observed between AU, AMR, and HAI incidence represent population-level correlations, not direct causal relationships. Ecological analyses are useful for identifying large-scale temporal patterns and informing stewardship priorities, but they cannot determine whether antibiotic exposure in individual patients directly caused the observed resistance or infection trends. Factors such as changes in case mix, infection control practices, diagnostic intensity, and surveillance methods may also influence the results. Consequently, while our findings highlight meaningful correlations and trends relevant for stewardship and infection prevention, they should be interpreted with caution and complemented by patient-level studies to establish causality and quantify individual risk.

This study has several more limitations. It was conducted in a single center, limiting broader comparability. Indications for AU (prophylaxis vs. treatment, community vs. hospital-acquired infections) were not systematically captured. Antimicrobials initiated prior to ICU admission were excluded. Additionally, our results reflect aggregate antibiotic use and may mask important patient-level differences, such as whether high DOTs stem from a few patients receiving prolonged therapy, many receiving shorter courses, or both. We also did not distinguish between monotherapy and combination therapy. Therefore, we cannot determine which prescribing patterns may contribute most to the development of AMR.

Nonetheless, strengths include prospective collection of data over 7 years, inclusion of more than 2000 ICU patients, and comprehensive analysis of HAI epidemiology, AU, microbiological etiology, and AMR. To our knowledge, this is the first ICU-based study in Serbia to address these issues in such depth.

## 4. Materials and Methods

The Military Medical Academy (MMA) in Belgrade, Serbia, a teaching hospital affiliated with the University of Defense, is a 1000-bed tertiary healthcare center divided into 27 departments according to medical specialty. The Clinic for Anesthesiology and Critical Care includes a 24-bed surgical ICU. Data collection in the ICU was conducted by the unit’s staff in collaboration with the Department for HAI Prevention and Control, which oversaw epidemiological surveillance. An infection control nurse participated in daily ICU rounds and completed standardized questionnaires for each patient. All collected data were subsequently entered into a dedicated software database (SPSS version 26.0., Chicago, IL, USA) developed for this purpose.

We prospectively collected surveillance data on all patients hospitalized in the ICU for more than 48 h, regardless of whether they acquired an HAI, including information on risk factors. These critically ill patients were referred for monitoring, observation, and management from various surgical specialties, including general surgery, neurosurgery, urology, traumatology, cardiac surgery, and thoracic surgery.

The following variables were recorded: patient characteristics (gender, age, transfer from another hospital, infection at admission to the ICU, diagnosis according to the International Statistical Classification of Diseases and Related Health Problems, Tenth Revision [ICD-10], and outcome of treatment [survived/deceased]); healthcare-related factors in the ICU (length of stay, presence of CVC, UC, MV, and surgery); and AU data for each patient. Antimicrobial indication per 100 “treatment days” or DOTs, as well as the ID of use for each antimicrobial group expressed in DOTs per 100 patient-days, were estimated. Antibiotics were classified as “Access,” “Watch,” or “Reserve” according to the 2019 WHO AWaRe Classification Database [60]. Prescriptions of systemic antifungals (ATC code J02) were also recorded. Antiviral agents (J05) were not included in this analysis, as the study focused on antibacterial and antifungal use patterns in relation to resistance and infection trends. Post-discharge surveillance was not performed.

A patient was defined as having a HAI if pneumonia, bloodstream infection, UTI [61], or HA-CDI [62,63] was acquired during ICU stay. All clinical specimens obtained from patients with HAIs were processed according to the routine diagnostic protocols of the Institute of Microbiology, MMA. Species-level identification was performed using matrix-assisted laser desorption/ionization time-of-flight mass spectrometry (MALDI-TOF; VITEK MS, bioMérieux, Marcy l’Etoile, France, Myla V4.9.2 VMS software V3.2 VMS ID Knowledge Base (IVD) V3.3). Since 2021, phenotypic detection of carbapenemase production has been performed using the NG-Test CARBA 5 immunochromatographic assay (NG Biotech, Guipry, France, V9.04.4). Antimicrobial susceptibility testing (AST) was carried out using the standard disk diffusion method in accordance with the European Committee on Antimicrobial Susceptibility Testing (EUCAST) guidelines. Minimum inhibitory concentrations (MICs) were determined using the Vitek2 Compact automated system (bioMérieux, Marcy l’Etoile, France). Colistin MIC determination was performed using the commercial broth microdilution method ComASP^®^ Colistin 0.25–16 (Liofilchem, Roseto degli Abruzzi, Italy). Quality control procedures followed the respective manufacturers’ instructions. Susceptibility to selected AMR markers was interpreted according to bacterial species as follows: *Staphylococcus aureus*: oxacillin (OXA), glycopeptides (GLY); *Enterococcus* spp.: GLY; *Enterobacterales* (*Escherichia coli*, *Klebsiella* spp., *Enterobacter* spp., *Proteus* spp., *Citrobacter* spp., *Serratia* spp., *Morganella* spp., *Providencia* spp.): third-generation cephalosporins (C3G: cefotaxime [CTX], ceftriaxone [CRO], ceftazidime [CAZ]); carbapenems (CAR: imipenem [IPM], meropenem [MEM], ertapenem [ETP]); *Pseudomonas aeruginosa*: CAR (IPM, MEM); *Acinetobacter* spp.: CAR (IPM, MEM).

For the detection of *Clostridioides difficile*, liquid stool samples from patients with suspected CDI (≥3 unformed stools within 24 h) were tested for glutamate dehydrogenase (GDH) and toxins A/B using the VIDAS^®^ enzyme immunoassay (bioMérieux, Marcy l’Etoile, France, V-1.4.4). GDH-positive and toxin-negative/indeterminate samples were analyzed by real-time PCR targeting the tcdB gene on the GeneXpert^®^ *C. difficile* system (Cepheid, Sunnyvale, CA, USA, V6.5).

Throughout the study period, standard IPC measures were in place in our ICU, including hand hygiene compliance, contact precautions for patients with MDR organisms, environmental cleaning and disinfection protocols. During the COVID-19 pandemic (2020–2022), infection control practices were further intensified. These included: universal masking and use of full personal protective equipment (PPE) for all patient contacts, isolation of patients with confirmed or suspected SARS-CoV-2 infection, enhanced environmental disinfection, and temporary reductions in elective admissions, which altered patient mix and antimicrobial exposure patterns.

Statistical analyses were performed using SPSS, version 26.0 (SPSS Inc., Chicago, IL, USA). The statistical unit for all analyses was the annual aggregate ICU data, with rates expressed per 1000 patient-days (or per 10,000 patient-days for HA-CDI). Antibiotic use was expressed as days of therapy (DOT) per 100 patient-days.

Results were presented as absolute numbers and percentages, or as means with standard deviations (SD) or medians with interquartile range (IQR), depending on data distribution (Kolmogorov–Smirnov test). Correlations between antibiotic use, incidence densities of HAIs, and antimicrobial resistance were assessed using Spearman’s rank correlation coefficient (ρ). Trends in antibiotic use and infection rates were analyzed by simple linear regression, and the coefficient of determination (R^2^) and regression slope (β) were reported to quantify the strength and direction of association. The Mann–Kendall trend test was additionally applied to confirm monotonic trends in antibiotic consumption and related variables. A *p* value < 0.05 was considered statistically significant. Figures were generated in Microsoft Excel (Microsoft Corp., Redmond, WA, USA). All statistical symbols used in tables and figures are defined as follows: ρ—Spearman correlation coefficient; R^2^—coefficient of determination; *p*—probability value; ID—incidence density; AU—antibiotic use; HAI—healthcare-associated infection; HA-CDI—hospital-acquired Clostridioides difficile infection.

Finally, an analysis of direct medical costs of antibiotic consumption was conducted from the perspective of the Republic Health Insurance Fund. Prices were obtained from the official price lists published annually by the Agency for Medicines and Medical Devices of Serbia and are expressed in Serbian dinars (RSD).

## 5. Conclusions

This seven-year ecological study provides one of the most comprehensive assessments of antibiotic utilization, healthcare-associated infections, and antimicrobial resistance in a Serbian surgical ICU. Although total antibiotic consumption remained overall stable, we observed a concerning shift toward increased use of Reserve antibiotics, reflecting the growing burden of multidrug-resistant pathogens. At the same time, the incidence density of healthcare-associated infections—particularly *Clostridioides difficile* infection—rose markedly. The resistance profiles of key Gram-negative pathogens, most notably *Klebsiella pneumoniae*, revealed an alarming escalation of carbapenem and colistin resistance, posing critical therapeutic and infection control challenges. These findings underscore the urgent need for reinforced antimicrobial stewardship programs, continuous surveillance of healthcare-associated infections, and strict infection prevention strategies. Future efforts should focus on optimizing the use of WHO “Access” antibiotics, restricting last-resort agents to well-defined indications, and investing in rapid diagnostic tools, infection prevention resources, and staff education. A strong multidisciplinary approach is essential to curb antimicrobial resistance, improve patient outcomes, and ensure the sustainability of critical care in Serbia and comparable healthcare settings.

## Figures and Tables

**Figure 1 antibiotics-14-01110-f001:**
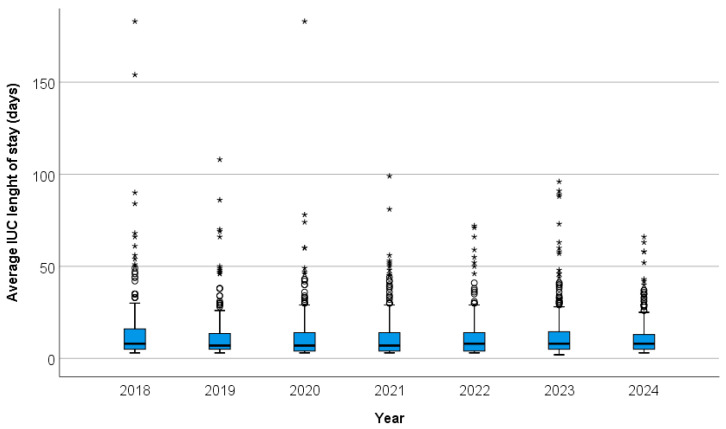
Average ICU length of stay in days during 2018–2024. The circle is an indication that an outlier is present in the data. The asterisk (*) is an indication that an extreme outlier is present in the data.

**Figure 2 antibiotics-14-01110-f002:**
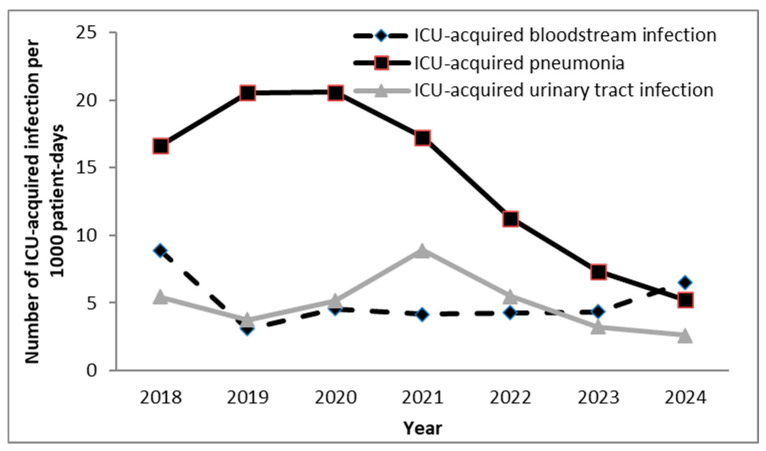
Incidence density of ICU-acquired infections per 1000 patient-days in surgical ICU the tertiary university hospital in Serbia. ICU—intensive care unit.

**Figure 3 antibiotics-14-01110-f003:**
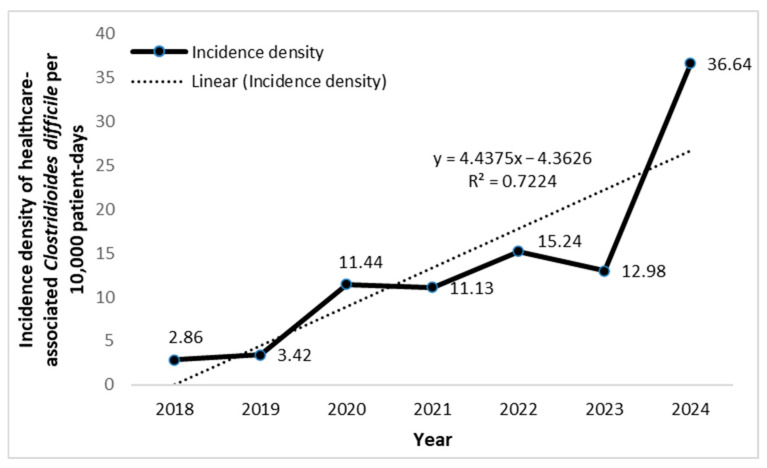
Incidence density (per 10,000 patient-days) of healthcare-associated *Clostridioides difficile* in the surgical ICU of a tertiary university hospital in Serbia.

**Figure 4 antibiotics-14-01110-f004:**
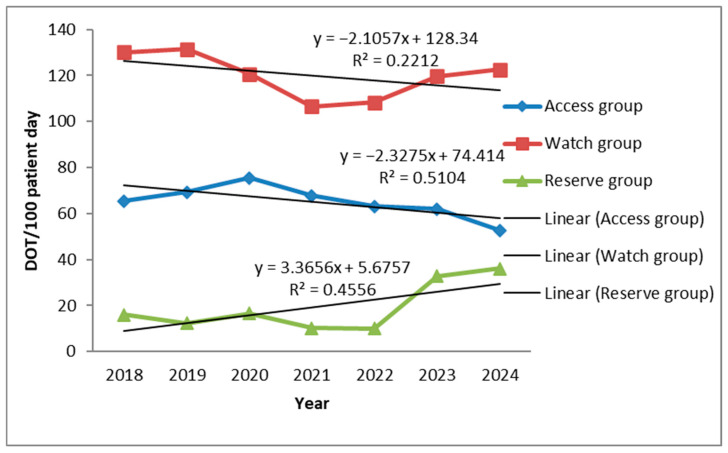
Trend of antibiotic consumption in days of therapy (DOT) per 100 bed-days according to WHO Access Watch Reserve Classification.

**Figure 5 antibiotics-14-01110-f005:**
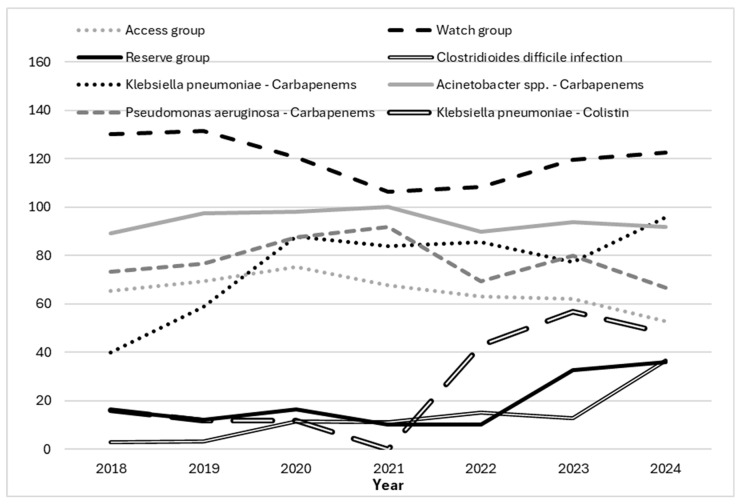
Trends in Antibiotic Consumption, Carbapenem Resistance, and *Clostridioides difficile* Infection in the ICU, 2018–2024. Y-axis: antibiotic use (DOT/100 patient-days), CDI incidence (per 10,000 patient-days), and resistance rates (%) for key ICU pathogens.

**Figure 6 antibiotics-14-01110-f006:**
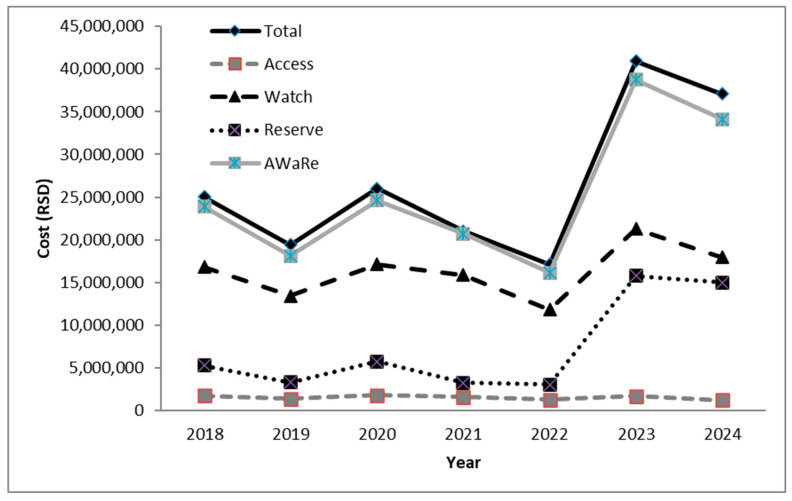
Assessment of the economic burden on the ICU due to the use of antibiotics (according to WHO Access Watch Reserve Classification). RSD—Serbian dinars; AWaRe—Access Watch Reserve.

**Table 1 antibiotics-14-01110-t001:** Characteristics of 2055 patients treated in the surgical ICU over a 7-year period (2018–2024).

	Year								
Characteristics; N (%)	2018 N = 238	2019 N = 236	2020 N = 298	2021 N = 298	2022 N = 279	2023 N = 367	2024 N = 339	Total N = 2055	*p*
Male sex	159 66.8	163 69.1	179 60.1	181 65.6	183 65.6	216 58.9	209 61.7	1290 62.8	0.081
Age	62.38 ± 17.272	63.10 ± 16.536	65.20 ± 15.941	64.48 ± 16.422	63.91 ± 16.601	65.68 ± 116.145	64.06 ± 16.090	64.25 ± 16.397	0.128
Age ≥ 65	116 48.7	130 55.1	178 59.7	173 58.1	154 55.2	220 60.1	191 56.3	1162 56.6	0.128
Surgery	136 57.1	157 66.5	189 63.4	187 62.8	190 68.1	263 71.7	234 69.0	1356 66.0	0.023
Transfer from other hospital	13 5.5	15 6.3	52 17.4	50 16.8	42 15.1	25 6.8	19 5.6	220 10.7	1.000
Infection on admission to ICU	40 16.8	62 26.3	71 23.8	70 23.5	72 25.8	107 29.2	66 19.8	453 22.0	0.704
Central vascular catheter	222 93.3	227 96.2	275 92.3	276 92.6	256 91.8	357 97.3	317 93.5	1930 93.9	0.254
Urinary catheter	196 82.4	233 98.7	293 98.3	296 99.3	276 98.9	366 97.8	337 99.4	1997 97.2	0.128
Mechanical ventilation	217 91.2	196 83.1	237 79.5	230 77.2	223 79.9	296 80.7	270 79.6	1669 81.2	0.610
Length of ICU stay (in days) (M-IQR)	8 (5−16)	7 (5−13.7)	7 (4−14)	7 (4−14)	8 (4−14)	8 (5−15)	8 (5−13)	/	0.180
In hospital mortality	110 55.9	108 45.8	130 43.6	149 50.0	143 51.3	146 39.8	154 45.4	940 45.7	0.704
In hospital mortality with HAI (%)	47 56.9	36 49.3	56 62.9	48 59.3	41 71.9	40 54.8	39 72.2	307 60.1	0.128
In hospital mortality without HAI (%)	63 41.2	72 44.2	74 35.4	101 46.5	102 45.9	106 36.1	115 40.4	633 41.0	1.000

M-IQR—medians with interquartile range; HAI—healthcare-associated infections.

**Table 2 antibiotics-14-01110-t002:** Trends of antibiotic utilization over 7-year period in the surgical ICU of tertiary university hospital in Serbia according to WHO Access, Watch, Reserve Classification.

ATC Code	DOT Per 100 Patient-Days							
Year	2018	2019	2020	2021	2022	2023	2024	*p* *
Total consumption of antibiotics	240.39	236.77	224.10	190.60	190.28	242.39	242.03	1.000
J01GB06 Amikacin	0.000	6.197	8.862	11.043	9.052	8.916	7.223	0.134
J01CR02 Amoxicilline Clavulanic acid	0.057	0.000	0.000	0.334	0.000	0.000	0.000	/
J01CA01 Ampicillin	0.000	0.000	0.000	0.000	0.000	0.000	0.000	/
J01CR01 Ampicillin Sulbactam	0.000	0.240	0.000	0.000	0.000	0.000	0.000	/
J01CE09 Procaine benzylpenicillin	0.000	0.000	0.000	0.000	0.000	0.000	0.000	/
J01CE01 PenicillinGsodium	0.000	0.000	0.000	0.000	0.000	0.000	0.000	/
J01DB04 Cefazolin	1.318	0.205	1.801	1.725	0.000	0.628	1.649	1.000
J01BA01 Chloramphenicol	0.000	0.000	0.000	0.000	0.000	0.000	0.000	/
J01FF01 Clindamycin	1.060	1.917	0.114	0.501	1.128	3.636	4.030	0.057
J01GB03 Gentamicin	1.031	1.609	0.829	0.779	0.518	2.402	0.366	0.128
J01XD01 Metronidazole	59.353	57.994	62.550	50.320	48.766	42.783	37.163	0.002
J01EE01 Sulfa.trimeto	2.635	1.198	1.229	3.032	3.596	3.614	2.329	0.128
J01FA10 Azithromycin	0.086	0.000	1.744	0.334	0.000	0.000	0.000	/
J01DC02 Cefuroxime	12.117	6.231	16.810	12.656	14.904	9.089	10.364	1.000
J01DD01 Cefotaxime	0.286	0.000	0.000	0.000	0.000	0.974	0.183	/
J01DD09 Ceftazidime	1.604	0.890	0.343	0.834	0.640	0.281	0.262	0.002
J01DD04 Ceftriaxone	7.992	24.444	8.262	12.295	15.087	16.447	19.969	0.057
J01DE01 Cefepime	3.552	2.396	1.429	0.890	2.073	2.986	0.968	0.254
J01MA02 Ciprofloxacin	0.201	0.205	0.343	1.558	0.000	0.173	1.361	0.617
J01FA01 Erythromycin	0.000	0.000	0.000	0.000	0.000	0.000	0.000	/
J01DH03 Ertapenem	4.555	3.458	9.920	9.930	8.412	4.977	3.559	1.000
J01DH51 Imipenem Cilastatin	20.682	14.071	6.575	13.046	12.923	11.534	11.515	0.023
J01MA12 Levofloxacin	5.586	5.272	3.316	3.310	4.115	4.090	2.329	0.023
J01DH02 Meropenem	28.588	27.764	33.533	22.142	19.994	32.417	34.284	0.704
J01MA14 Moxifloxacin	0.000	0.000	0.000	0.195	0.000	0.000	0.000	/
J01CR05 Piperacillin Tazobactam	8.021	8.456	3.087	0.612	1.372	5.908	5.234	0.447
J01XA02 Teicoplanin	3.351	0.000	0.000	0.000	0.000	0.000	0.000	/
J01XA01 Vancomycin	33.543	38.206	35.363	28.734	28.741	30.859	32.478	0.447
J01XB01 Colistimethate sodium	9.940	8.216	12.579	7.121	7.802	18.589	20.073	0.254
J01XX08 Linezolid	3.753	3.184	1.630	2.253	1.402	10.755	12.248	0.704
J01AA12 Tigecycline	2.378	0.924	2.373	0.751	0.884	1.060	0.497	0.057
J01DD52 Ceftazidime–avibactam	0.000	0.000	0.000	0.000	0.000	2.337	3.219	/
J02AC01 Flukanazol	28.702	22.937	10.978	6.092	7.802	27.440	28.710	1.000
J02AC03 Vorikonazol	0.000	0.342	0.000	0.111	0.488	0.000	0.000	/
J02AC04 Posakonazol	0.000	0.000	0.000	0.000	0.000	0.000	0.000	/
J02AX04 Kaspofungin	0.000	0.308	0.429	0.000	0.579	0.195	1.910	/
A07AA02 Nystatin	0.000	0.000	0.000	0.000	0.000	0.000	0.105	/
J02AX05 Mikafungin	0.000	0.000	0.000	0.000	0.000	0.303	0.000	/
J04AB02 Rifampicin	0.000	0.103	0.000	0.000	0.000	0.000	0.000	/
ACCESS GROUP	65.454	69.360	75.385	67.734	63.060	61.979	52.760	0.023
WATCH GROUP	130.164	131.496	120.725	106.536	108.261	119.735	122.506	0.447
RESERVE GROUP	16.071	12.324	16.582	10.125	10.088	32.741	36.037	0.447
Antifungal medicines	28.702	23.587	11.407	6.203	8.869	27.938	30.725	1.000

*—Mann–Kendall test, DOT—days of therapy.

**Table 3 antibiotics-14-01110-t003:** Correlation between consumption of particular antibiotics and incidence density of ICU acquired *Clostridioides difficile* infection in the tertiary university hospital in Serbia, 2018–2024.

ATC Code	Antibiotics	Total ICU ID of *Clostridioides difficile **
J01MA02	Ciprofloxacin	r = −0.071 *p* = 0.879
J01DD04	Ceftriaxone	r = 0.393 *p* = 0.383
J01XD01	Metronidazole	r = −0.75 *p* = 0.052
J01XA01	Vancomycin	r = −0.393 *p* = 0.383
J01MA12	Levofloxacin	r = −0.643 *p* = 0.119
J01DH02	Meropenem	r = 0.286 *p* = 0.535
J01XB01	Colistimethate Na	r = 0.429 *p* = 0.337
J01MA14	Moxifloxacin	r = −0.204 *p* = 0.661
J01GB03	Gentamicin	r = −0.571 *p* = 0.180
J01DE01	Cefepime	r = −0.429 *p* = 0.337
J01CR01	Ampicillin + sulbactam	r = −0.408 *p* = 0.363
J01DH03	Ertapenem	r = 0.071 *p* = 0.879
J01AA12	Tigecycline	r = −0.607 *p* = 0.148
J01FA10	Azithromycin	r = −0.433 *p* = 0.331
J01CR05	Piperacillin + tazobactam	r = −0.429 *p* = 0.337
J01FF01	Clindamycin	r = 0.500 *p* = 0.253
J01DB04	Cefazolin	r = −0.071 *p* = 0.879
J01XX08	Linezolid	r = 0.107 *p* = 0.819
J01DC03	Cefuroxime	r = 0.143 *p* = 0.760
J01EE01	Trimethoprim + Sulfamethoxazole	r = 0.321 *p* = 0.482
J01DH51	Imipenem + cilastatin	r = −0.750 *p* = 0.052
J01XA02	Teicoplanin	r = −0.612 *p* = 0.144
J01DD09	Ceftazidime	r = −0.893 *p* = 0.007
J01GB06	Amikacin	r = 0.429 *p* = 0.337
J01DD01	Cefotaxime	r = 0.059 *p* = 0.900
J01CR02	Amoxicilline + clavulanic acid	r = −0.579 *p* = 0.173

*—Spearmen’s correlation.

**Table 4 antibiotics-14-01110-t004:** Selected antimicrobial resistance markers of *Klebsiella pneumonia*, *Providenttia stuarti*, *Acinetobacter* spp., *Pseudmonas areuginsa*, *Staphyloccous aureus* and *Enterococcus* spp. in the tertiary university hospital in Serbia, 2018–2024.

	Resistance (%)							
Microorganism	Antibiotic	2018	2019	2020	2021	2022	2023	2024
*Klebsiella pneumoniae*	3G	21/25 84.0%	18/17 94.01%	25/25 100.0%	23/25 92.0%	26/28 92.9%	40/44 90.9%	23/23 100%
	Carbapenems	10/25 40.0%	10/17 58.8%	22/25 88.0%	21/25 84.0%	24/28 85.7%	34/44 77.27%	22/23 95.65%
	Colistin	4/25 16.0%	2/17 11.76%	3/25 12.0%	0/25 0%	12/28 42.86%	25/44 56.82%	11/23 47.83%
*Acinetobacter* spp.	Carbapenems	41/46 89.1%	40/41 97.56%	51/52 98.07%	45/45 100.0%	27/30 90.0%	15/16 93.75%	11/12 91.67%
	Colistin	14/46 30.43%	1/41 2.44%	0/52 0%	0/45 0%	0/30 0%	2/16 12.5%	0/12 0%
*Pseudomonas aeruginosa*	Carbapenems	11/15 73.33%	13/17 76.47%	28/32 87.5%	11/12 91.67%	9/13 69.23%	12/15 80.0%	6/9 66.67%
	Colistin	2/15 13.33%	0/17 0%	1/32 3.12%	0/12 0%	2/13 15.38%	2/15 13.33%	0/9 0%
*Providentia stuartii*	3G	3/5 60.0%	3/3 100.0%	1/1 100.0%	4/4 100.0%	2/2 100.0%	1/1 100.0%	5/7 71.43%
	Carbapenems	0/5 0.0%	1/3 33.3	0/1 0.0	3/4 75.0%	1/2 50.0%	1/1 100.0%	5/7 71.43
	Colistin	0/5 0%	0/3 0%	0/1 0%	0/4 0%	1/2 50.0%	0/1 0%	0/7 0%
*Staphylococcus aureus*	Oxacylin	10/11 90.9%	6/9 66.67%	2/4 50.0%	6/6 100.0%	6/7 85.71%	0/2 0%	0/0 0%
*Enterococcus* spp.	Vancomycin	4/7 57.1%	1/4 25.0%	5/8 62.5%	3/8 37.5%	0/3 0%	4/8 50.0%	4/10 40.0%

**Table 5 antibiotics-14-01110-t005:** Trend of selected antimicrobial resistance markers and correlation between consumption of particular antibiotics and AMR of ICU-acquired HAI in the tertiary university hospital in Serbia, 2018–2024.

	Antimicrobial Resistance		Antibiotic Consumption		Correlations **	
Microorganism	Resistance	Trend *	Antibiotic	Trend *	Coefficient	*p* Value
*Klebsiella* spp.	3rd Cephalosporin	0.254	3rd Ceph	0.128	−0.214	0.645
	Carbapenems	0.057	Carbapenems	0.447	−0.107	0.819
	Colistin	0.046	Colistin	0.254	0.714	0.071
*Acinetobacter* spp.	Carbapenems	1.000	Carbapenems	0.447	−0.250	0.589
	Colistin	/	Colistin	0.254	0.177	0.704
*Pseudomonas aeruginosa*	Carbapenems	0.704	Carbapenems	0.447	−0.071	0.879
	Colistin	/	Colistin	0.254	−0.037	0.937
*Providentia stuartii*	3G	1.000	3rd Ceph	0.128	0.089	0.849
	Carbapenems	/	Carbapenems	0.447	−0.523	0.229
	Colistin	/	Colistin	0.254	−0.408	0.363
*Staphylococcus aureus*	Oxacylin	/	Vancomycin	0.447	−0.306	0.504
*Enterococcus* spp.	Vancomycin	0.617	Vancomycin	0.447	0.357	0.432

*—Mann–Kendall test, **—Spearmen’s correlation.

**Table 6 antibiotics-14-01110-t006:** Comparison of ICU-Acquired Infection Rates: Present Study vs. ECDC vs. INICC.

Infection Type	Present Study (Serbia, 2018–2024)	ECDC Mean (2019–2021) [1,33,34]	INICC (2015–2020) [4]
Pneumonia/VAP	14.1/20.0 per 1000 patient-/ventilator-days	6.8 per 1000 patient-days	11.96 per 1000 ventilator-days
Bloodstream Infection (BSI)	5.1 per 1000 patient-days	5.1 per 1000 patient-days	—
CVC-related BSI (CLABSI)	2.3 per 1000 CVC-days	—	4.55 per 1000 central line-days
Urinary Tract Infection (UTI/CAUTI)	4.9/5.1 per 1000 patient-/catheter-days	2.9 per 1000 patient-days	2.91 per 1000 catheter-days

## Data Availability

The data sets used and/or analyzed in this present study are available from the corresponding author upon reasonable request.

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
