# Peer review of "Antibiotic Use, Healthcare-Associated Infections, and Antimicrobial Resistance in Intensive Care Unit of a Serbian Tertiary University Hospital, 2018–2024: An Ecological Analysis"

_antibiotics, 2025, doi:10.3390/antibiotics14111110_

Round 1

Reviewer 1 Report

Comments and Suggestions for Authors
  1. Were antifungal and antiviral agents included in the surveillance, or was the analysis restricted to antibacterial drugs?
  2. How were HAIs diagnosed—strictly by EU/ECDC criteria, or were clinical diagnoses also included?
  3. What infection control measures were in place during the study period, and did these change during COVID-19?
  4. Was any molecular typing attempted for pneumoniae or Acinetobacter isolates (for example to detect carbapenemase genes)?
  5. Could the mortality rates reported be stratified by infection status (patients with vs. without HAIs)?
  6. Add clear, integrated visualizations showing overall antibiotic consumption trends, AMR evolution, and CDI incidence, which would allow readers to quickly grasp the relationships between these variables.
  7. Expand comparisons with data from ECDC and other global surveillance systems, highlighting how Serbia’s trends align with or diverge from other European countries.
  8. Discuss antibiotic costs not only as absolute figures but also relative to ICU budgets or national healthcare expenditures, and link this to potential cost savings from stewardship interventions.
  9. Explicitly note how ecological data differ from patient-level analyses, and caution against inferring causality between AU and resistance/HAI trends.

Author Response

We would like to express our gratitude to the reviewers for their insightful comments and careful evaluation of our work. Their suggestions have helped us significantly refine and strengthen the manuscript. All requested revisions have been implemented, and the modifications are highlighted in yellow throughout the text.

Comment 1. Were antifungal and antiviral agents included in the surveillance, or was the analysis restricted to antibacterial drugs? 

We thank the reviewer for this observation and the opportunity to clarify the scope of antimicrobial surveillance. As we stated in the Materials and Methods “Prescriptions of systemic antifungals (ATC code J02) were also recorded.”

The analysis included systemic antibacterial (J01) and antifungal (J02) agents, while antiviral drugs (J05) were not part of the dataset. This approach was chosen because the surveillance system in our ICU routinely monitors antibacterial and antifungal use, whereas antiviral therapy is recorded separately and was outside the focus of this study. We have clarified this in the revised version of the Methods section.

Comment 2. How were HAIs diagnosed—strictly by EU/ECDC criteria, or were clinical diagnoses also included? 

We thank the reviewer for this important question. All healthcare-associated infections (HAIs) were diagnosed strictly according to the European Centre for Disease Prevention and Control (ECDC) surveillance definitions and diagnostic criteria.

Comment 3: What infection control measures were in place during the study period, and did these change during COVID-19? 

We thank the reviewer for this relevant question. Throughout the study period, standard infection prevention and control (IPC) measures were in place in our ICU, including hand hygiene compliance, contact precautions for patients with multidrug-resistant organisms, environmental cleaning and disinfection protocols. During the COVID-19 pandemic (2020–2022), infection control practices were further intensified. These included: universal masking and use of full personal protective equipment (PPE) for all patient contacts, isolation of patients with confirmed or suspected SARS-CoV-2 infection, enhanced environmental disinfection, and temporary reductions in elective admissions, which altered patient mix and antimicrobial exposure patterns.

We have clarified this in the revised version of the Materials and Methods section.

Comment 4: Was any molecular typing attempted for K.pneumoniae or Acinetobacter isolates (for example to detect carbapenemase genes)? 

We appreciate the reviewers’ valuable observations and interest in molecular typing. The revised Discussion now clarifies that our conclusions are based on phenotypic resistance testing, complemented by carbapenemase detection using the NG-Test CARBA 5 assay. We also added a reference to our recent molecular study (Rakonjac B, Djurić M, Djurić-Petković D, Dabić J, Simonović M, Milić M, Arsović A. Evaluation of the NG-Test CARBA 5 for Rapid Detection of Carbapenemases in Clinical Isolates of Klebsiella pneumoniae. Antibiotics. 2025; 14(10):989. https://doi.org/10.3390/antibiotics14100989), which demonstrated a 99.4% genotypic resistance rate of K.pneumoniae to carbapenems using PCR. This addition provides molecular context for our findings and outlines the direction for future research on genetic determinants of antimicrobial resistance. While detailed genotypic data for Acinetobacter isolates are not yet available, this analysis is currently in progress within our laboratory. We plan to include these results in a follow-up study focused specifically on the molecular epidemiology of carbapenem-resistant Acinetobacter spp. in our ICU.

Comment 5: Could the mortality rates reported be stratified by infection status (patients with vs. without HAIs)? 

We thank the reviewer for this helpful suggestion. In the revised manuscript, mortality rates have been stratified according to infection status. Two new rows (“In-hospital mortality with HAI” and “In-hospital mortality without HAI”) were added to Table 1. Additionally, we included a new reference (Rosenthal VD, Yin R, Lu Y, Rodrigues C, Myatra SN, Kharbanda M, Valderrama-Beltran SL, Mehta Y, Daboor MA, Todi SK, Aguirre-Avalos G, Guclu E, Gan CS, Jiménez-Alvarez LF, Chawla R, Hlinkova S, Arjun R, Agha HM, Zuniga-Chavarria MA, Davaadagva N, Basri MNM, Gomez-Nieto K, Aguilar-de-Moros D, Tai CW, Sassoe-Gonzalez A, Aguilar-Moreno LA, Sandhu K, Janc J, Aleman-Bocanegra MC, Yildizdas D, Cano-Medina YA, Villegas-Mota MI, Omar AA, Duszynska W, BelKebir S, El-Kholy AA, Alkhawaja SA, Florin GH, Medeiros EA, Tao L, Memish ZA, Jin Z. The impact of healthcare-associated infections on mortality in ICU: A prospective study in Asia, Africa, Eastern Europe, Latin America, and the Middle East. Am J Infect Control. 2023 Jun;51(6):675-682. doi: 10.1016/j.ajic.2022.08.024.) supporting the association between HAI and increased mortality among ICU patients.

Comment 6: Add clear, integrated visualizations showing overall antibiotic consumption trends, AMR evolution, and CDI incidence, which would allow readers to quickly grasp the relationships between these variables.

We thank the reviewer for this helpful suggestion. In response, we have added integrated visualizations illustrating overall antibiotic consumption (Access, Watch, and Reserve groups), the evolution of antimicrobial resistance among key ICU pathogens, and the incidence density od HA CDI). These figure (now Figure 4) allow readers to quickly appreciate temporal relationships between antibiotic use, resistance trends, and HA CDI dynamics.

Comment 7: Expand comparisons with data from ECDC and other global surveillance systems, highlighting how Serbia’s trends align with or diverge from other European countries. 

We thank the reviewer for this valuable suggestion. In the revised version, we have expanded the comparative analysis to include data from the European Centre for Disease Prevention and Control (ECDC) and the International Nosocomial Infection Control Consortium (INICC). New results comparing infection incidence densities across systems have been incorporated into both the Results and Discussion sections (see Table 6 and the corresponding text).

Specifically, our ICU’s mean rates for pneumonia, bloodstream infection, and urinary tract infection were compared with ECDC averages (expressed per 1,000 patient-days), while device-associated rates for VAP, CLABSI, and CAUTI were benchmarked against INICC pooled data (per 1,000 device-days).

We have also explicitly noted the methodological differences between surveillance systems (patient-days vs device-days) to prevent misinterpretation of numerical comparisons.

Revisions made:

  1. New subsection in Results presenting comparative rates and updated text with two additions: “when expressed per MV days, the mean rate of ventilator-associated pneumonia (VAP) was 20.0 per 1,000 MV-days” and “when expressed per urinary catheter days, the mean rate of catheter-associated urinary tract infection (CAUTI) was 5.1 per 1,000 UC-days”
  1. Added paragraph in Discussion interpreting alignment and divergence from ECDC and INICC data and Table 6 with data of comparations.

Comment 8. Discuss antibiotic costs not only as absolute figures but also relative to ICU budgets or national healthcare expenditures, and link this to potential cost savings from stewardship interventions.

We appreciate this valuable suggestion. In the revised manuscript, we expanded the discussion of antibiotic costs to include their proportion relative to the total health expenditure in Serbia. The updated text now highlights that antibiotic consumption in our ICU represented approximately 0.005–0.007% of national healthcare spending—indicating that, while antibiotic costs constitute a small share of total expenditures, stewardship interventions can result in significant proportional savings.

We have also added references to international evidence demonstrating that antimicrobial stewardship programs (ASPs) can reduce antibiotic costs by 20–35% without compromising patient outcomes:

(Karanika S, Paudel S, Grigoras C, Kalbasi A, Mylonakis E. Systematic Review and Meta-analysis of Clinical and Eco-nomic Outcomes from the Implementation of Hospital-Based Antimicrobial Stewardship Programs. Antimicrob Agents Chemother. 2016 Jul 22;60(8):4840-52. doi: 10.1128/AAC.00825-16.

Baur D, Gladstone BP, Burkert F, Carrara E, Foschi F, Döbele S, Tacconelli E. Effect of antibiotic stewardship on the in-cidence of infection and colonisation with antibiotic-resistant bacteria and Clostridium difficile infection: a systematic review and meta-analysis. Lancet Infect Dis. 2017 Sep;17(9):990-1001. doi: 10.1016/S1473-3099(17)30325-0.).

These revisions explicitly connect antibiotic use patterns with cost-effectiveness and the potential economic impact of stewardship interventions.

Comment 9. Explicitly note how ecological data differ from patient-level analyses, and caution against inferring causality between AU and resistance/HAI trends. Vesna

We appreciate this important methodological observation. In the revised manuscript, we added a new paragraph in the Discussion explicitly clarifying that our study is based on ecological (aggregate) surveillance data rather than patient-level data.

The revised text now emphasizes that the observed correlations between antibiotic use, resistance, and infection rates do not imply causality. We also note that factors such as case mix, diagnostic intensity, and infection control measures may confound these relationships. The paragraph concludes by recommending patient-level analyses to confirm causality and further quantify individual risk.

Reviewer 2 Report

Comments and Suggestions for Authors
  1. The AMR characterization (genetypic) was not described and it limits molecular interpretation of the data.
  2. Moreover, as this manuscript related with ecological and descriptive study, there is limitations in casual inference. It would be better to add this information and discuss at the introduction and discussion section.
  3. The statistic unit and annotations were not described. It is suggested to strength the statistic work.
  4. Please also improve figure legends for clarity. Table 1 contains redundant ICD-10 details, and it will be better to move to Supplementary Material.
  5. At the discussion section descriptive results was described instead of interpreting them mechanistically.

Author Response

We would like to express our gratitude to the reviewers for their insightful comments and careful evaluation of our work. Their suggestions have helped us significantly refine and strengthen the manuscript. All requested revisions have been implemented, and the modifications are highlighted in yellow throughout the text.

Comment 1: The AMR characterization (genotypic) was not described and it limits molecular interpretation of the data.

We are grateful for this comment. In the revised manuscript, we have augmented the Discussion to include not only that genotypic testing (PCR) was previously performed, but which carbapenemase genes predominated in our region (predominance of OXA-48-like, followed by NDM and KPC). This addition provides mechanistic specificity and grounds our phenotypic findings within a regional molecular context.

Comment 2: Moreover, as this manuscript related with ecological and descriptive study, there is limitations in casual inference. It would be better to add this information and discuss at the introduction and discussion section.

We appreciate this important methodological observation. In the revised manuscript, we added a new paragraph in the Discussion explicitly clarifying that our study is based on ecological (aggregate) surveillance data rather than patient-level data. The revised text now emphasizes that the observed correlations between antibiotic use, resistance, and infection rates do not imply causality. We also note that factors such as case mix, diagnostic intensity, and infection control measures may confound these relationships. The paragraph concludes by recommending patient-level analyses to confirm causality and further quantify individual risk.

Comment 3: The statistic unit and annotations were not described. It is suggested to strength the statistic work.

We thank the reviewer for this helpful observation. In the revised manuscript, we clarified the statistical unit (annual ICU-level aggregate data expressed per 1,000 patient-days or per 10,000 patient-days for HA-CDI) and expanded the description of statistical methods in the Materials and Methods section. We also specified the tests used (Spearman’s correlation, linear regression, and Mann–Kendall trend test) and added definitions for all statistical symbols (ρ, R², p) used in tables and figures. These additions strengthen the transparency and reproducibility of the statistical analysis.

Comment 4: Please also improve figure legends for clarity.

We have revised and expanded all figure legends to improve clarity and readability. Each legend now provides a concise explanation of the variables presented, units of measurement, and abbreviations used (e.g., antibiotic groups according to the WHO AWaRe classification, and infection types). Specifically, the legend for Figure X has been updated to clearly describe the temporal trends in antibiotic consumption, carbapenem and colistine resistance among key ICU pathogens, and HA CDI incidence from 2018–2024.

Comment 5: Table 1 contains redundant ICD-10 details, and it will be better to move to Supplementary Material.

We thank the reviewer for this helpful suggestion. As advised, the ICD-10 diagnostic details previously included in Table 1 have now been moved to the Supplementary Material to improve clarity and focus of the main text. Table 1 in the revised manuscript now presents only the key demographic and clinical characteristics of patients, while the full list of ICD-10 diagnoses can be found in Supplementary Table S1.

Comment 6: At the discussion section descriptive results was described instead of interpreting them mechanistically.

We thank the reviewer for this constructive observation. In the revised manuscript, we expanded the Discussion section to include a mechanistic interpretation of antibiotic use trends. Specifically, a new paragraph was added, hypothesizing how the absence of active stewardship oversight, changes in patient mix, and heightened infection control practices during the COVID-19 period may have influenced prescribing behavior. The revised text also considers broader institutional and behavioral factors shaping antibiotic use in critical care, thereby providing a deeper analytical interpretation consistent with the reviewer’s recommendation.

Round 2

Reviewer 2 Report

Comments and Suggestions for Authors

Manuscript can be accepted for publication.